# Causal inference in ethnographic research: Refining explanations with abductive logic, strength of evidence assessments, and graphical models

**Jeffrey G. Snodgrass**[1]\*, **H. J. François Dengah, II**[2], **Seth I. Sagstetter**[1], **Katya Xinyi Zhao**[1]

**1** Department of Anthropology and Geography, Colorado State University, Fort Collins, Colorado, United States of America, **2** Department of Sociology and Anthropology, Utah State University, Logan, Utah, United States of America

\* jeffrey.snodgrass@colostate.edu

**Data Availability Statement:** All relevant data are within the article and its Supporting Information files.

## Abstract

In their classic accounts, anthropological ethnographers developed causal arguments for how specific sociocultural structures and processes shaped human thought, behavior, and experience in particular settings. Despite this history, many contemporary ethnographers avoid establishing in their work direct causal relationships between key variables in the way that, for example, quantitative research relying on experimental or longitudinal data might. As a result, ethnographers in anthropology and other fields have not advanced understandings of how to derive causal explanations from their data, which contrasts with a vibrant "causal revolution" unfolding in the broader social and behavioral sciences. Given this gap in understanding, we aim in the current article to clarify the potential ethnography has for illuminating causal processes related to the cultural influence on human knowledge and practice. We do so by drawing on our ongoing mixed methods ethnographic study of games, play, and avatar identities. In our ethnographic illustrations, we clarify points often left unsaid in both classic anthropological ethnographies and in more contemporary interdisciplinary theorizing on qualitative research methodologies. More specifically, we argue that for ethnographic studies to illuminate causal processes, it is helpful, first, to state the implicit strengths and logic of ethnography and, second, to connect ethnographic practice more fully to now well-developed interdisciplinary approaches to causal inference. In relation to the first point, we highlight the *abductive* inferential logic of ethnography. Regarding the second point, we connect the ethnographic logic of abduction to what Judea Pearl has called the *ladder of causality*, where moving from *association* to *intervention* to what he calls *counterfactual* reasoning produces stronger evidence for causal processes. Further, we show how graphical modeling approaches to causal explanation can help ethnographers clarify their thinking. Overall, we offer an alternative vision of ethnography, which contrasts, but nevertheless remains consistent with, currently more dominant *interpretive* approaches.

**Funding:** The Foundation for Psychocultural Research (https://thefpr.org/). "Online gaming involvement, avatar identification, and emotion regulation in five culture areas: A multi-level cultural norm and social network approach." P.I.: Jeffrey G. Snodgrass. The funders had no role in study design, data collection and analysis, decision to publish, or preparation of the manuscript.

**Competing interests:** The authors have declared that no competing interests exist.

## Introduction

In their classic accounts, anthropological ethnographers develop causal arguments for how specific sociocultural structures and processes shape human thought, behavior, and experience in particular settings. For example, Bronislaw Malinowski's research in the Trobriand Islands illuminated how social status concerns influenced kula ring exchange patterns [1]. Marjorie Shostak showed how !Kung beliefs about gender roles shaped practices related to sexuality, childbirth, marriage, family structure, and livelihoods [2]. Colin Turnbull's ethnography points to how Mbuti subsistence strategies partially determined this group's spiritual beliefs and practices [3]. Margaret Mead argued that Samoan childrearing and socialization practices impacted adolescent and adult sexuality [4]. E.E. Evans-Pritchard showed how fission and fusion within and between Nuer lineage groups shaped how domestic conflicts and even wars were resolved [5] and also how Zande social tensions and accompanying emotions produced witchcraft beliefs and practices [6]. Oscar Lewis famously (and controversially) argued that a culture of poverty contributed to ongoing social and economic distress in Mexico [7]. Closer to the present, Phillippe Bourgois and Paul Farmer demonstrated how structural inequalities and racism promote ongoing disease and suffering throughout the world [8, 9].

Despite the prominence of causal explanation in classic ethnographies, contemporary researchers influenced by Geertz's "thick description" tradition of ethnography [10], a still dominant perspective in anthropology, generally aim to provide deep, contextually rich, and largely qualitative descriptions of social phenomena. That is, they *interpret* culture and society —and ideally at least in part through a local "native's point of view"—rather than *causally explain* how one factor shapes another. In this view, ethnography is presented as not well-suited to establishing direct causal relationships between key variables in the way that, for example, quantitative research relying on experimental or longitudinal data might. After all, naturally occurring sociocultural contexts are dynamic, complex, and messy, where multiple factors interact with each other to shape human thought and behavior. And ethnographic methods like participant-observation and qualitative interviews do not lend themselves to isolating or examining how specific key variables might directly impact others in ways necessary to establish causation. As Geertz (p. 316) himself put it,

> As interworked systems of construable signs (what, ignoring provincial usages, I would call symbols), culture is not a power, something to which social events, behaviors, institutions, or processes can be causally attributed; it is a context, something within which they can be intelligibly—that is, thickly—described [10].

In contrast to such cultural anthropological suspicion toward causal analysis, researchers in the broader qualitative social sciences, and particularly in the discipline of sociology, have advanced understandings of how to derive causal explanations from largely qualitative data and analysis. For example, historical sociologists have developed *qualitative comparative analysis* to clarify how complex events like revolutions result from specific combinations of factors among different nations [11]. Other sociologists have developed *analytic induction* to move from *how* to *why* explanations [12–14]. This method involves developing tentative causal claims from the analysis of qualitative data like fieldnotes and interview transcripts. Those claims are then substantiated or revised through a *constant comparative method* with other *counterfactual* cases where a given presumed causal factor under somewhat different conditions might be associated with a similar or different outcome [13–16]. *Theoretical sampling* of next cases is critical in the manner is allows researchers to select cases that allow them to test their evolving causal claims. Purporting to proceed by *induction* in the conventional sense, the

method is used to analytically reduce individual cases into overarching categories that capture key features and relationships within the data, much like grounded theory approaches [17, 18]. Finally, others in sociology and related fields are more interested in illuminating causal mechanisms and thus answering *how* questions [19]. These researchers are less interested in whether changes in a first factor are associated with changes in a second and more in how that change is brought about. Researchers working from this perspective have been particularly interested to understand how individuals respond to or make meaning from changing circumstances, which in turn shapes how subsequent changes unfold [20, 21].

Nevertheless, it has been noted by a prominent sociologist with extensive ethnographic experience: "Few scholars in the aforementioned traditions have asked what role *ethnographic* data might play in causal analysis. In addition, few *ethnographers* have addressed these contemporary causal perspectives in any depth" [emphasis ours] [22]. Given this gap in understanding, we aim in the current article to clarify the potential ethnography has for illuminating causal processes related to the cultural influence on human thought, behavior, and experience, thus providing an anthropological counterpoint to more dominant interpretive perspectives. In doing so, we want to clarify points often left unsaid in both classic anthropological ethnographies and in more contemporary interdisciplinary theorizing on qualitative research methodology about how exactly *ethnography* might yield insights into causal processes. We argue that for ethnographic studies to do this effectively, it is helpful, first, to state the implicit strengths and logic of ethnography and, second, to connect ethnographic practice more fully to now well-developed interdisciplinary approaches to causal inference.

In relation to the first point, we highlight in the current article the largely *abductive* inferential logic of ethnography. Faced with puzzling field observations, ethnographers *abduce* or *retroduce* (in Peirce's terminology [23]) a theoretical proposition that would explain the puzzle. Following related research in anthropology and sociology, we argue that logical abduction as a creative process—something like informed guessing—is central both to the practice of ethnography and to how ethnographers develop causal theories to explain the world [24–27]. Of note, our focus on abduction is different than a more typical understanding of ethnography as centrally inductive, as in grounded theory approaches [17, 18, 28], though induction and also deduction can help ethnographers to further develop and validate causal theories, as we also show.

Regarding the second point, we connect the ethnographic logic of abductive inference to what Judea Pearl has called the *ladder of causality*, where moving from *association* to *intervention* to what he calls *counterfactual* reasoning produces stronger evidence for causal processes. Ethnographers, we show, can observe and document each of Pearl's causal ladder rungs. Further, we also show how graphical modeling approaches to causal explanation, as seen, for example, in so-called "directed acyclic graphs" or "DAGs" developed by Pearl and others can help ethnographers clarify their causal thinking [29–32]. This leads us to a distinctive approach to qualitative data analysis, where ethnographers aim to identify and code causal relations in qualitative data, and thereby construct ethnographically informed graphical causal models. Such models include not only causal relationships of primary interest to researchers, but also how other factors might, for example, *confound* or *mediate* causal relationships between a predictor and an outcome. Such formal modeling is characteristic of epidemiology and many social science disciplines such as economics, but connecting as we do here that form of modeling to ethnographic inference and qualitative analysis is not standard. Finally, we show how the ethnographically informed graphical causal models can be subsequently tested with quantitative data, such as that collected in field surveys. We consider the latter analysis, though formal and quantitative, to be thoroughly *ethnographic*, given its grounding in understandings

built up through ongoing field observations and interviews and engagement with culturally informed and contextually sensitive theory.

In developing these arguments, we speak, first, to ethnographers who want to improve their causal explanations and logic and thus also the strength and validity of their conclusions. As we see it, ethnographers' sometimes *implicit* causal arguments—like how poverty and social marginality in Brazil change maternal affect [33, 34]–could be further strengthened through *explicit* attention to the principles we present. We also articulate, second, how ethnographers' understandings of culturally inflected experiences and "native points of view" are critical to developing sound causal arguments. Anthropology's paradigm wars sometimes pit more humanistic, critical, and interpretive forms of ethnography against scientific and naturalistic approaches aiming for causal explanation [35, 36]. However, our approach can also help interpretivists better *explain* cultural processes, like how underlying culturally informed motives drive distinctive behaviors [37], thus pointing to synergies between ethnographic interpretation and explanation.

## Background theory: Causal inference in ethnographic research

### The abductive logic of ethnographic research

Ethnography is a case study account of the cultural knowledge and behavior characteristic of a particular group or setting [1, 38, 39]. The "culture" piece relates to the fact that humans as a social species have the evolved ability to think and behave jointly or collectively [40–42]. That is, humans transmit information to each other, which becomes the basis of collective attention, intention, and action [43–45]. This is important because it implies that humans are incomplete without such socially learned information and behavior [10, 46]. Methodologically, this means that to fully explain human behavior in a specific setting researchers would need to understand local culture—that is, patterns of socially learned thought and practice [43]. And ethnographers developed techniques to do just that. Those methods included qualitative modes of inquiry such as participant-observation and interviews, which typically included open-ended phases of study to reveal among other things unexpected patterns that were not yet known to the researcher [47]. However, as an ethnographer's knowledge of a particular place and culture grows, they might include more structured and even quantitative methods like field surveys to further clarify local thinking and behavior [47–51].

Ethnography, then, is more a particular mode of inquiry and logic than any specific method or technique, be it qualitative or quantitative, open-ended or structured [24]. Key to our understanding of ethnographic method is the (often unrecognized but ever present) role of logical abduction [23–25, 28]. In Peirce's writing, abductive inference can be characterized as follows:

The surprising fact C is observed.

But if A were true, C would be a matter of course.

Hence, there is reason to suspect that A is true [23, 27].

Novelty, then, is critical to ethnographic inquiry [24]. An ethnographer is confronted with surprising observations in the field that cannot be easily explained according to current understandings of the world. These are surprising because they are based on socially learned structures of knowledge and practice that are not yet known to the ethnographer. Even ethnographers originating from a given culture group would still not understand *all* dimensions of this *particular* form of thinking and practice in this *specific* setting. The ethnographer subsequently draws on whatever resources available—their experience, current theory—to identify some A that would make C a matter of course. As Peirce noted, this form of logical inference—what he called informed guessing—relies on human creativity and invention [23].

Importantly, compared to both induction and deduction (discussed later), "A is neither assumed before the fact (as it is in deduction) nor observed (as it ideally is in induction). Rather, the proposition is guessed at, presumed after the fact to explain observations we cannot easily explain away" [27].

As explanations invented after the actual occurrence of an event, which was not actually witnessed, abductive inferences are neither rigorously true in a logical sense (as they are in deduction) nor necessarily statistically common (as they are in induction). Rather than representing a logical weakness per se, Peirce instead thought of abduction as the most creative of human logical processes [23]. Further, he thought, as do we, that abduced explanations can be further refined and complemented via inductive and deductive logic. This has been described as abduction's *iterative* and *recursive* logic [24]. Ethnographers repeat (iterate) this form of logic until arriving at a satisfying conclusion. And prior explanations become embedded in earlier ones (recursion). As we will show, logical induction is important here: for an abduced explanation to be compelling, ethnographers should see repeated data points consistent with the proposed inference. Deduction also has its place: to gain greater confidence in an abduced explanation, ethnographers can test whether such explanations lead ethnographers to deduce subsequent propositions from the ethnographic findings that are logically consistent with each other.

To summarize, abduced explanations, as invented, can originate from various places. That could be the experience of the ethnographer, which, we would add, suggests that socially and culturally diverse ethnographic teams, such as those combining relative insiders and outsiders to a given setting, would have access to a greater reservoir of potential explanations. Or, alternately, those explanations could come from current theory, though necessarily adapted in some form to the new cultural circumstances. This has also been conceived of as a strength of abductive logic: in contrast to "inductive" approaches like grounded theory, abductive explanations can more effectively build upon and extend current theoretical understandings characteristic of a community of researchers [28, 52].

## Assessing strength of evidence via the ladder of causality: Association, intervention, counterfactuals

Another piece of our argument, besides recognizing the relevance of logical abduction, is what Judea Pearl refers to as a three-level *ladder of causality*, with each level or rung of the ladder identifying a stronger causal process [29, 53]. The lowest level, *association*, is linked with seeing and observing regularities or patterns in the world. In statistics, one object is associated with another if observing the first object changes the probability of seeing the second one. Pearl notes how association alone is weakly indicative of causality in such instances, given a first object might be the cause of a second, the second might cause the first, or a third object or process (referred to as a *confounder*) might cause both of the two to occur. The middle level is *intervention*, which refers to how actions produce changes in the world. Experiments are the classic example of this level, and random assignment to the treatment or control group helps to ensure that the treatment, rather than some other non-random characteristic of the individuals in the two groups, produces the change. Ethnographers are more likely to observe naturally occurring events—which we refer to as *interventions*, though typically without conscious design—*that change the value of a predictor variable in ways that can impact a study's outcomes*. The highest *counterfactual* level or rung of the ladder involves building theories of the world that explain why specific factors like a treatment have effects and what might happen in the absence of those factors [54, 55]. This can involve thinking through how different versions of past events (that were not observed) could have produced alternate versions of the present.

In quantitative analysis, this means building models that allow a researcher to set key variables (such as a mediator) at different levels and observe how other parts of the model change, like how a main predictor is associated with an outcome.

In the current study, we show how ethnography can help reveal processes linked to each of Pearl's causal levels. Ethnographers commonly identify associations between different factors at their field sites. Those linkages can help clarify causal processes. Ethnographers also observe naturally occurring events in the field that result in other things happening. Their careful observation of these events—naturally occurring experiments or *interventions* of a kind—can reveal how those events result in the changes. Here, it is critical to document, as we discuss next, how other factors at the field site might confound or mediate how the process being observed unfolds. Finally, ethnographers can work in a comparative counterfactual mode where they can observe and document how slightly different social and personal situations (established at some point in the past) have led to alternate present circumstances. This is challenging, and documenting the role played by confounders, mediators, and the like is again critical. Having respondents reflect on their experiences in the different situations and what might have produced them can also help ethnographers build more valid theories to explain the causal processes at play.

## Ethnographically refining and testing causal understandings: The role of graphical causal models (DAGs)

Social science researchers have developed a language and approach for representing causal assumptions graphically, based in part on foundational work by Sewell Wright [56, 57]. One class of these graphical causal models are referred to as *directed acyclic graphs* (DAGs). In epidemiological and social science applications of such graphs, study variables are represented as *nodes* or *vertices* and their causal connections as directed *arrows*, *edges*, or *paths* that move in specific directions, as seen for example in Fig 1. The graphs are "acyclic" in the sense that following the directed edges will never result in a closed loop. Bidirectional causality can be represented in DAGs by creating predictor or outcome vertices at different timepoints. Fig 1 presents in abstract terms a hypothetical research project's primary causal assumption that the study Predictor influences the Outcome. Further, the Predictor might also impact the Mediator, also shown in Fig 1, which in turn influences the Outcome. Such an intermediary variable might help researchers further explain the mechanism that connects the Predictor to the Outcome. Other causally prior factors that influence both the Predictor and the Outcome are referred to as Confounders. And causally downstream variables that can open illusory causal associations between the Predictor and the Outcome—because they occur after the causal process of concern—are referred to as Colliders. In general, such graphical models help researchers determine the identifiability of causal effects from observed data (i.e., whether the data includes all relevant factors). And they direct them on how to treat various classes of study factors. In reference to the latter, social science researchers would generally: examine how the Predictor-Outcome relationship changes under various Mediator values or conditions, to clarify that potential causal mechanism [29, 58]; account or "control" for Confounding variables (covariates) that might bias an estimated association between the Predictor and Outcome and thus create *spurious* causal relationships; and be wary of Colliders that if accounted for might deceptively magnify the purported causal association between the Predictor and Outcome [29, 30, 32].

Of note, DAGs are *qualitative* models of how the world works: they only encode, for example, that there is a directed causal relationship between a predictor and an outcome; they do not say anything about the magnitude or functional form of that causal relationship. As

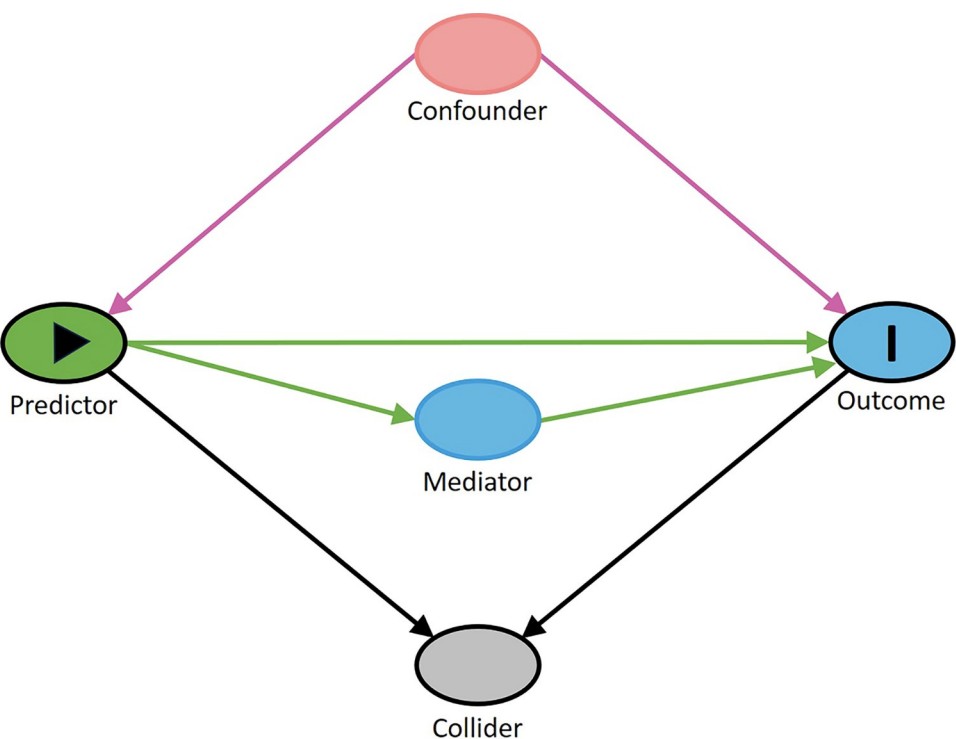

**Fig 1. A graphical causal model expressed as a DAG.**

qualitative models, DAGs are highly amenable to being refined through qualitative modes of ethnographic analysis of qualitative data (like using abductive or other forms of logic to code fieldnotes and interview transcripts). (Here, we follow others in using qualitative and quantitative to refer either to modes of analysis or to forms of data [59, 60].) But qualitative understandings of the world derived ethnographically and represented in DAGs can also be further refined and tested via quantitative analysis of quantitative data. For example, based on preliminary work, an ethnographer might develop and distribute a field survey to test (via regression analysis or some other strategy) causal processes encoded in an ethnographically derived DAG. This would form a multiphase research strategy consisting of exploratory and confirmatory stages of study. Nevertheless, to our mind, the project would remain thoroughly ethnographic if the derived understandings and concerns were all closely tied to culturally informed experiences, thinking, and behavior characteristic of the field site. All in all, graphical causal models provide ethnographers with a powerful tool to represent abstractly their evolving understandings (iteratively and recursively abduced) about what influences what at their specific study site. Representing their understandings in this way provides other researchers with a more transparent window into an ethnographic account, as a reflection of ethnographers' expert knowledge and beliefs but also potential biases. And rendering causal assumptions more visible (to themselves and to other researchers) clarifies what is unique (or not) about a particular field site, which helps to advance theory. (For more detail on graphical modeling in relation to interdisciplinary work on causal inference, see S1 Appendix).

## Methods and approach

To illustrate these points on causal explanation in ethnographic research, we draw from our lab's ongoing studies of culture, gaming, and psychosocial well-being. For over a decade now,

our research lab has been ethnographically documenting (via observations, interviews, and field surveys) how video gaming is linked to players' overall experience of well-being in their lives. That research has been largely focused on massively multiplayer online roleplaying games (MMORPGs or MMOs) like *World of Warcraft*, *Final Fantasy XIV*, and *Guild Wars 2*, though with attention to how MMO player experiences differ from those associated with other games such as single player RPGs (e.g., *Skyrim*), massive online battle arenas or MOBAs (such as *League of Legends* and *Dota 2*), first- and third-person shooters (like *Call of Duty*), and fighting games (including *Super Smash Bros*). In that earlier research, we ethnographically and psychometrically validated positive and negative gaming experience scales, which we employ in the current study [61].

More recently, we have aimed to clarify how gamers' *avatar* experiences differ in roleplaying games (RPGs) as compared to other forms of play, and how such differences can causally shape gaming-related well-being. That more recent research has included 100s of hours of in-game observations, over 100 semi-structured interviews, and more than 500 field surveys. It also has meant that our research team now extensively documents avatar and character experiences in games played face-to-face on tabletops (TTRPGs such as *Dungeons & Dragons* or *D&D*) or enacted live and in-person (so-called live-action roleplaying games or LARPs). (For a recent synthesis of this earlier work, see [62]).

The research described in this article, including the use of appropriate informed consent procedures, has been reviewed and approved by the Colorado State University Institutional Review Board (CSU IRB) for the protection of human subjects (protocol #2634). Recruitment for the research presented here began November 1st, 2021 and ended May 15th, 2023. For our interviews and observations, we relied on verbal consent, as that both minimized the chance that participants might be identified and was more practicable and appropriate for our ethnographic study where building rapport and trust with study participants in fluid and dynamic naturalistic contexts is so critical. This meant that we first explained our research to study participants—and provided respondents with supporting documentation—and then proceeded with the research only after receiving oral confirmation that respondents consented to participate. For our surveys, we followed the same verbal consent procedure when recruiting respondents. Then, the surveys themselves repeated that information, and by proceeding with the survey participants signaled that they consented to participate in our study. The CSU IRB approved these consent procedures.

## Ethnographic findings

### Ethnography's abductive, iterative, and recursive logic in relation to gaming and player well-being

Early on in our ethnographic research, we learned that some players of the video game *World of Warcraft* (WoW) claimed to be *addicted* to that game. Such reports were surprising to members of our research team who had only recently begun playing that game, as they had not considered that a game could be experienced in this way. In part, reporting to be addicted to WoW revealed that a player found a game well-constructed, compelling, and fun. But it was more than that for certain players, who also found themselves playing compulsively in ways that eroded commitments to their families, partners and spouses, schooling, and jobs. Reporting WoW "addiction" was also a way of communicating the distress that this game created in their lives.

Our ethnographic observations and interviews showed us how important WoW play was to some players, which helped to explain its addictive qualities. As we advanced in our study of WoW and other games, we began to see how gaming avatars or characters could be akin to

secondary or proxy selves for some players. That is, games were motivating in part because players experienced themselves as personally present in some powerful way in the games they were playing. A character's challenges, victories, and defeats, then, were the players' as well. And many players could not easily separate themselves from their characters. To put it more strongly, we learned how digital gaming avatars allowed players to enact and experience in virtual game-worlds their best and most ideal selves. WoW guild members, for example, developed in online characters more highly accomplished selves, which provided them with elevated social standing in this virtual world and guild setting. And those avatars, as idealized second or proxy selves, had the potential to help players compensate for their perceived life failings, thereby bolstering their psychosocial well-being.

The existing scholarship partially guided us throughout this newer phase of ethnographic study. In one well-known early formulation, Bessière and colleagues spoke of how gaming avatars or characters were akin to *ideal elves* (in the sense of a digital gaming *pixie* or *elf*) [63]. Other researchers investigating avatar processes in video games and other digital environments have used Edward Higgins's self-discrepancy theory (SDT) framework to help explain well-being in these contexts [64]. First formulating his ideas in the late 1980s, Higgins, a health psychologist, identified three separate aspects of the self: what he called the *actual self* (the traits one actually possesses), the *ideal self* (the traits one would ideally like to possess), and the *ought self* (the traits someone believes one should or ought to possess) [65, 66]. Research in the SDT tradition shows that discrepancies between an individual's perception of their actual and ideal selves are associated with dejection, sadness, disappointment, and depression, whereas discrepancies between perceived actual and ought selves are linked to agitation-related states such as anxiety, nervousness, and guilt. Adapted for virtual worlds research, the relevant self-dimensions are the actual self (defined in this context as the characteristics of the offline user), the ideal self (the traits one would optimally possess), and the *avatar self* (characteristics of the user's online personae). A recent review of the literature suggests that the avatar self is typically created as "better than the user, resembling the ideal self more closely than the actual self" [64]. Even more relevant to what we were learning, a now substantial body of research suggests that gamers suffering from low self-esteem—predicted by SDT as resulting from felt tensions emerging from the perceived disjunction between the actual and ideal self [67]—tend to create avatars even more closely resembling their ideal selves, possibly in order to compensate for perceived shortcomings and thereby bolster psychosocial well-being. These adaptations of SDT, then, helped us to clarify what we were learning ethnographically about avatar therapeutic mechanisms: as approximations of individuals' ideal selves, avatars provide opportunities for the enactment of one's best self.

We might summarize these findings in the following proposition:

Proposition 1. *Some gamers are highly motivated to play—even describing their gaming as "addictive"—because their character-avatars come to feel akin to self-objects or self-proxies, even ideal selves, which can help psychosocially compensate players for their felt personal deficiencies.*

This idea was logically abduced, in Peirce's terms [23], in the sense that what was once puzzling to us (WoW players' deeply motivated and even compulsive and "addictive" game play) is clarified. The explanation is causal because a felt experience—my avatar-character is in some important sense also *me*—explains how gaming can come to play therapeutic (or in other situations distressing) roles in players' lives. Proposition 1 was unexpected to us, but once we identified it, puzzles we encountered in the field (like addictive game play) became a matter of

course. Of note, in an abductive approach, we allowed ourselves to make use of any ideas—be they from our own observations, interviews, and experiences or from the extant scholarly literature—that helped explain the puzzles we encountered in the field. This meant that we were able to engage current scholarship in ways not possible by approaches like grounded theory that aim to be more purely inductive. Our findings did echo and thus confirm in important ways the research of Bessière, Mancini, and others.

As we pushed further ahead in our inquiries, we learned that players in fact could relate differently to their avatar-characters. Some did relate to them as self-objects or proxies—like a *me*—as we described previously. But others treated them like distinctive *other persons* separate from themselves. And still others used them more impersonally like a *tool* or *object*. In relation to the latter, the lead author still remembers vividly an early interviewee correcting his assumption that this player's relationship to his avatar was personal and deeply felt. Instead, the interviewee said that he used his gaming character more like he would a chess or backgammon piece—that is, as an object or tool to accomplish his in-game goals.

As in the previous example described in this section, there is literature on this topic, though we learned about them later. For example, research shows how video game players can relate to gaming avatars as *object*, *me*, *symbiote*, and *other* [68]. Each of these vary in terms of how players perceive their emotional investment in avatars, the autonomy of avatars from players themselves, the suspension of disbelief in relation to avatars, and their relative degree of control over avatars [69]. Experiencing an avatar as an *object* is associated with low emotional investment, low avatar autonomy (as players feel that digital avatars simply do players' bidding), low suspension of disbelief (avatars are not imagined to be anything other than a digital tool), and a high degree of felt control. By contrast, feeling that the avatar is the self or *me* would be linked to high emotional investment, little avatar autonomy (the player and avatar are felt to be one), a high degree of suspension of disbelief (to imagine that the avatar is in fact identical to the self), and less of a sense of external control (because the player and avatar are a single entity). Further, players feeling they have a *symbiotic* relationship with their avatars—that is, merged to some extent as a single being and yet still distinctive, so that players feel that they "bleed" into their characters [70]—are likely to show somewhat less emotional investment and suspension of disbelief relative to players feeling that the avatar is *me*, and somewhat higher avatar autonomy and felt control compared to *avatar is me* kinds of players. Finally, perceiving the avatar to be a distinctive being or *other* is associated with high degree of emotional investment (because players establish human-like relationships with avatars), high avatar autonomy (the avatar is separate from the player), high suspension of disbelief (to imagine that the avatar is a "real" person), and less of a degree of felt control (the avatar does what it wants).

Further, we began to understand ethnographically that the kinds of relationships players established with their characters could impact the positive or negative quality of their overall gaming experiences. We thought this because those four kinds of player-avatar relationships bring players into either more or less intimate and meaningful contact with their avatars, thus potentially magnifying both gaming-related well-being benefits and distress. Thus, for example, greater emotional investment in avatars would mean that players' achievement, social, and immersion experiences would be felt more keenly, thus potentially magnifying (for better or worse) the positive or negative consequences of play. In this sense, players relating to their avatars as *me* or as *symbiotes* (with high personal and emotional investment in those alter egos) would therefore more likely experience both greater gaming-related well-being gains and be at higher risk of problem play. Likewise, experiencing avatars as more autonomous would point to how players feel distinct from their avatars and thus more distanced or even protected in some sense from whatever achievement, social, or immersion experiences those gaming

avatar-characters have. But following that logic, players feeling that their avatars are a *me* or a *symbiote* (with less autonomy) would again more likely experience the highs (and lows) of gaming-related pleasure and distress. Further, a high suspension of disbelief in relation to avatars would again suggest more intensely felt achievement, social, and immersion gaming experiences, in the sense that what-if gaming worlds are felt to be *real* in some important way. This would again mean that players relating to their avatars as a *me* or a *symbiote* (and as *other*) would potentially experience both greater gaming-related pleasure and distress. Finally, feeling a higher degree of control over avatars could be experienced as quite pleasurable, especially for players who experienced little to no control over their actual-world lives (connected, for example, to achievement-related feelings of mastery). Such feelings could promote a pleasurable sense of *flow* [71] and immersive escape. Or, instead, they might lessen the felt independence and thus reality of imagined *others* in the game, which might instead reduce the felt pleasures of immersive role-playing.

We summarize these ideas as follows:

Proposition 2. *Players' relative gaming-related pleasure and distress emerges in part from their specific relationships to their character-avatars, with some relating to characters in highly personal terms as extensions of the self or alternately as separate living persons or as impersonal objects or tools that help one accomplish gaming goals.*

This second proposition (also logically abduced) helped us to explain new puzzles in the field. In this case, those puzzles related to differences we were noticing across gaming genres. For example, players of role-playing games—MMOs like WoW, but also tabletop roleplaying games (TTRPGs) like *Dungeons & Dragons* (D&D) and live-action roleplaying games or LARPs dramatically enacted in person—tended to relate to their characters as a *me* or *symbiote*. Players of other games like shooters or MOBAs, by contrast, instead related to avatars in more impersonal terms. And those differences also seemed to help us predict patterns in players' gaming-related well-being.

This second conclusion also points to the iterative and recursive quality of ethnographic research, which Agar highlights. We *iterated* or repeated our ethnographic observations and interviews—and our engagement with relevant scholarship—until we arrived at a new conclusion that we felt advanced our understanding of how and why gaming could be so motivating for some players. Further, our prior conclusion—Proposition 1—was *recursively* included in this second conclusion. Earlier conclusions about players' more personal relationship to their avatar-characters—as self-proxies or substitutes—are referenced in and even foundational to Proposition 2. However, in this case, current theory could only take us so far. The link between player-avatar relationships and gaming-related well-being had not yet been systematically addressed.

So, we set out to do that, as we describe in the next section. As readers will see, we present those findings in a different way. Specifically, we do so in relation to Pearl's ladder of causality, showing how different ethnographic observations and interviews allowed us to move up in causal certainty from association to intervention to counterfactuals. That analysis shares many features with what has been called *analytic induction* and *grounded theory* [15, 17]. Following that presentation, we illustrate how we presented our understanding of those causal processes in a graphical causal model or DAG, which was amenable to more formal testing via a field survey and quantitative analysis. Confirming our earlier observations and interviews in that way would give us even greater confidence in our findings.

### Strength of ethnographic evidence and the ladder of causality: Player-avatar relationships shape gaming-related well-being

**Associations.** The following ethnographic vignette illustrates how Cindy's relating to and investing in an avatar-character in highly personalized *me* terms (or perhaps as an ideal self *symbiote*) is associated with both emotional and well-being highs and lows: Cindy, 21, is relatively new to gaming, and joined an online RPG, in part, to try out a hobby that her younger brother was passionate about. She is also a student-collaborator on our research team [72]. While conducting participant-observation in the online roleplaying game *Guild Wars 2*, she records the following account in her journal, describing her experience creating the perfect character:

> It was a daunting task (creating my character). . . But in the end, I discovered that the effort I put into my avatar reflected a lot of different things. She reflected who I am, a bit of who I want to be, and what I want to be seen as in the virtual space as I conduct my research. I named her Nuwcindy. I made a second avatar because I wasn't sure if I was content with her as my character, as if perhaps she was too similar to myself, but when I played with my tiny little goblin-like creature it just didn't feel. . . . right.

The similarities between Cindy and Nuwcindy were more than just in appearance (and in name). Both Cindy and Nuwcindy are outgoing, bubbly, and friendly. It was no surprise, therefore, that early in her online experience, she (the two of them together?) wanted to make friends. She adopted a "new player" identity, asking other players for help and guidance as she began her ethnographic journey. It wasn't long before she was invited to join a guild, The Sinister Smurfs. Recounting the experience in her journal, Cindy wrote:

> I saw someone posting (in the chat) advertising for new and experienced players to join a guild. I responded to it. They were very friendly and said they were helping a lot of new players lately. . . I talked to them for a while and told them I was going to bed. Everyone said goodnight. I JOINED A GUILD!!!

For the next several weeks, Cindy would conduct participant-observation by playing with the Sinister Smurfs. The Smurfs would teach Cindy (and Nuwcindy) how to play the "game," and in turn, she would help them complete joint quests and adventures. All the while, she would listen to the members converse back and forth, and she would respond by typing in the guild chat (since Cindy didn't have a working microphone at the time). Cindy and the Sinister Smurfs (who happened to be several college-age male students living in Australia) would talk about school, their respective countries, and pop culture while vanquishing their enemies. They were interested that a student was conducting anthropological research on video games and were helpful in answering Cindy's questions. It wasn't long before that she started to consider these gamers as friends, even though they had never met face-to-face in the out-of-game world.

This all changed one day when Cindy typed, ". . .as a girl myself. . ." in response to a discussion about college dating. Immediately, the Sinister Smurfs, who had once been friendly to Cindy, became angry and combative. "WHAT? You are a GIRL? You lied to us!" and kicked her out of the guild without any further explanation. Cindy confessed that she felt "confused" and "hurt" by the events. "I guess they just assumed that I was a guy playing a girl character since so many do, I never meant to deceive them".

Cindy's experience shows the rewards and risks of relating to a character as *me* or *symbiote*. Nuwcindy was, in many ways, Cindy herself. Through her participant-observation, she created

an avatar that was an extension of herself into the virtual world. As such, she tried to create the social connections online that she values offline. She succeeded in finding a new group of friends that made her happy, which brought her back as Nuwcindy night after night. However, the sudden dissolution of these social bonds caused her a "surprising" amount of psychological pain, for what she thought were "just online" relationships.

It seems apparent that Cindy's highly personal way of relating to her character causally shapes her gaming-related well-being experiences. After all, it is in part because Cindy relates to her avatar in such highly personal ways (as a self-substitute) that her gaming feels so good and rewarding at first and then painful after she is removed from the guild. However, there are at least four *confounding* factors that complicate such a causal explanation: *gender*, *social support*, *gaming involvement*, and *gaming genre*. Readers will recall that a confounding variable influences both a predictor and an outcome variable, thus potentially rendering any causal association between them spurious. (See S1 Appendix for more detail.) In the first case, Cindy's gender is at the root of the gaming-related problems she experiences. That is, if she were male, she would not have experienced such unpleasantness. Her fieldnotes do not tell us if her gender also influenced the way she related to her character, though other ethnographic evidence of ours shows how female and non-binary players do in fact tend to relate to their characters in more personal *me* and *symbiote* terms. In the second instance, the root of Cindy's distress emerges from the gaming-related social support (or lack of support) she experiences. What is at first a highly supportive environment turns sour, as does Cindy's gaming experience. Though again not presented here, we do have other ethnographic examples that show how players only typically relate to their avatar-characters as self-proxies *when they feel safe and supported to do so by other players*. In relation to the third factor, incidents like this are typically associated with changes in gaming involvement, which complicate the causal picture. A once enthusiastic player like Cindy might reduce their play time (and emotional investment too) because of such incidents. This can help them minimize their gaming-related distress—both because they're playing less and because they're less likely to think of their characters as *me*—though the direct influence of player-avatar relationships on gaming-related well-being would need to account for these temporal processes. Finally, and fourth, we have found that playing role-playing games, as Cindy does, is associated both with specific player-avatar relationships (more likely to play a *me* or *symbiote*) and distinctive gaming highs and lows, thus again potentially confounding the causal process we aim to understand.

## Interventions

Naturally occurring events that that change the value of a predictor variable in ways that can impact a study's outcomes—or what we refer to as *interventions*—can provide firmer evidence (compared to mere associations) for causal processes. With over a decade of experience in *Dungeons and Dragons* (D&D), our respondent Ash's relationship to this game and their characters changed dramatically when they suddenly found themselves unemployed and decided to take up a career as a professional D&D game master (GM). This naturally occurring intervention (in Pearl's terms) moved Ash from relating to their characters in more personal *me* terms to less personal terms as *other persons* or *objects*, which in turn reduced some of the pleasures and psychosocial benefits they originally derived from gaming.

Ash described their early characters as follows:

I was very much like most people who get into it. I was very attached to the characters. I just enjoyed playing a character that spoke to me at that moment in time. Whatever I am feeling, whatever shows I've recently watched. Ideas and sparks. And I like exploring that.

In this early period, Ash used the medium of the tabletop roleplaying game (TTRPG) to explore less obvious aspects of their identity and to experience life how they imagined themselves or wanted to be. Later, however, pressures from their gaming group led Ash to begin gaming less and less often as a player and more as a game master, who was responsible for running and arbitrating games. If Ash didn't take up this role, they feared that his group would disband. Then, after five years of participating more casually in TTRPG play as both player and GM, Ash unexpectedly lost their job. To pay his bills, Ash decided to try his hand as a professional paid GM. By the time Ash was interviewed by a member of our research team, they were working online as a paid game host, helping to produce a D&D podcast, running their own online streaming channel, and guest starring in the podcasts and streams of other professional TTRPG content creators.

The transition from player to GM, and then from casual GM to professional player and GM, dramatically changed how Ash interacted with their own characters, as well as with other players and the game itself. The game had gone from, in Ash's words, a "hobby and passion and turned to essentially my job now." When talking about their characters after this transition, Ash described the new relationship as follows:

> I don't have any attachment to them anymore, because a lot of times those characters end up becoming *one shot* characters [i.e., temporary characters created for a particular gaming session or campaign]. Or it's a very small series where we're recording or streaming for, like at most ten episodes. So, I only get to be this character and in that character's shoes for at most three months. There's a disconnect as well, especially with *safety tools* and a lot of that stuff [rules that help protect players from experiences and content felt to be harmful]. I think that for me personally it's made that *bleed* happen less and less often [i.e., less blurring of the boundary between the player and character]. I am me, the player, and this is my character, almost a third person perspective. I am no longer like 'this character is me.'

Ash described the work of playing characters online in these professional groups as akin to putting on a theatrical production, where Ash was less able to invest emotionally in their characters and thus felt less connected to them.

By pursuing work as a guest player, meeting other content creators online, and inviting guests to their own shows, Ash also began playing with strangers. To accommodate the varied interests of players and their diverse backgrounds, professional GMs like Ash turn to safety tools to ensure they are not harming their players. Often safety tools involve players and GMs negotiating ahead of the game itself as to what topics and conflicts are acceptable in play. According to Ash, common topics of avoidance are racism, gender discrimination, and police brutality. As Ash mentioned above, the addition of safety tools contributed further to their feeling of emotional disconnection from their characters. Despite repeatedly mentioning in the interview that safety tools are an important part of their professional toolkit, it has also in part caused them to *bleed* less and less into the character in ways that earlier felt satisfying.

In those earlier periods, then, when playing casually, Ash was able to identify with their characters and pursue their personal passions through them. When their hobby became a profession and the game became work, Ash found themselves unable to immerse themselves in the game. The character was no longer *me*, but a "third person" or even an *object*. And the game was no longer relaxing and beneficial temporary escape from life's stresses, but work itself and thus one of the sources of those stresses.

In this vignette, circumstances push Ash to shift from relating to their characters as a *me* to a more distanced and impersonal "third person" or even *object* kind of relationship. And this shift made it more difficult for Ash to enjoy and benefit from the game in ways that earlier

came easily and naturally to them. These changes are brought about in part by what Pearl and others refer to as naturally occurring interventions. Members of Ash's early play group pressure Ash to take on the role of GM, without which the group might dissolve. In that role, Ash becomes a game arbitrator who still manipulates other characters, but they are largely *non-player characters*—or "NPCs"—who, as bit players, are largely there as props who help advance the story and drama of which the player-characters are stars. Ash became an expert at using those NPCs as tools—*objects*—to advance the game and story. Their relationship to the game's characters had changed, as had their original satisfaction playing their own characters. Then, financial pressures lead Ash to begin building a career as a professional GM. This furthers the erosion of earlier felt connection to their own characters, which was the foundation of how and why D&D was originally meaningful to them. Circumstances, then, *intervene* in ways that change Ash's relationship to their characters, which in turn erodes many of the pleasures and psychosocial benefits they originally derived from gaming.

According to Pearl and others, naturally occurring interventions like these provide stronger causal evidence because many potentially confounding factors are held constant before and after the interventions. Ash is still (roughly) the same person before and after these changes, and they're still playing the same game, in some cases with the same players in an identical gaming group. Ash, then, is their own statistical control, and this makes it more likely that the changes in their relationship to their characters—from *me* to *other* or *object*—*cause* the changes in the satisfaction they derive from gaming.

Of course, Ash's reality is more complicated than this. It is more accurate to say that Ash shifted their gaming role from that of *player* to *GM* to eventually *professional player and GM*. Central to those transitions, we would argue based on our ethnography, is a change in how Ash related to their characters. But other factors were also associated with that shift in roles. For Ash, these included greater time investment in D&D (they learned how being a GM is hard work), more responsibility, and in some instances new gaming groups, situations, and expectations. We cannot rule out the possibility that these other factors—bound up as they were with the changed manner Ash related with their characters—erode Ash's gaming pleasure and satisfaction. The *interventions* we describe, then, provide us with greater causal certainty compared to mere *associations*, because we can eliminate certain potential confounders as being causally determinant of gaming-related well-being in such situations. But the evidence is far from perfect, and we cannot rule out the potential importance of other factors. However, counterfactual evidence, as we show in the next section, can provide even greater degrees of causal certainty.

## Counterfactuals

Video games like *World of Warcraft* (WoW), where players inhabit online virtual worlds, provide players opportunities to create many different characters, with each crafted in ways that meet players' wants and game-related goals. Players of MMOs like WoW often have their primary character—called a *main*—often treated like an alter ego or *me-symbiote*. And they have secondary characters—*alts*—manipulate in ways that allow for exploration of other identity dimensions or to achieve particular gaming demands. The latter might include, for example, competitive player-vs-player mini-games inside WoW, amassing virtual resources (*farming* them) that help players achieve their goals (say, making *potions* that enhance players' *armor* so they can collaborate with others in *dungeons* or *raids* where the aim is to defeat potent enemies called *bosses*), or simply serving as a *mule* that holds a players' gaming property (serving as a kind of gaming *bank*). Online RPG players frequently—but not always—relate to their alts more as impersonal *tools* or *objects* that help them accomplish specific gaming tasks.

Most WoW players, then, have deep experience playing a variety of mains and alts. This allowed members of our research team to learn from players (via interviews and observations) how their relationships with their various characters might differentially shape the satisfaction and benefits they derived from WoW. Of particular interest, players we knew would typically settle on a main character, with whom they often had a deep and personal connection. And based on players' prior experiences, they could (on our prompting) imaginatively contrast how playing that character was inherently different in terms of well-being outcomes compared to *if* that character were somehow different. Such imaginative *what if* explorations allowed us to contrast players' *actual* current experiences with what Pearl refers to as counterfactual realities, which did not actually occur but could have. According to Pearl, such comparisons provide an even higher degree of causal certainty because the comparisons between what did actually occur and what instead might have occurred allow for a degree of precision in the isolation and manipulation of the predictor in relation to the outcome not possible in naturally occurring experiments or interventions.

Thus, take for instance the experiences of another respondent, Sam, an avid 24-year-old gamer, who played another video role-playing game called *Neverwinter Nights*. Sam plays a variety of computer role-playing games, regularly clocking in more than 60 hours a week. Until recently, Sam played predominantly male characters. Sam, however, had trouble relating with these male avatars. "Trying to connect with my male characters is hard. . . [I feel] disconnected from them." It wasn't until Sam realized that she was transgender that she was able to form a closer bond to her avatars by making them female like herself.

> Because of my recent life choice, one that I've been debating on for a long time. . .I finally ended up saying 'this is what I wanna be.' Just being able to say that. Like my whole mindset for how my characters are or how my character should be ended up completely changing. It was because of this choice that made me think that maybe I want to be a female in these games, because maybe they would make me feel like I'm actually in the game.

Creating female avatars, and in some cases, changing those characters' gender from male to female, allowed Sam to connect with them in a deeper, more meaningful way, and to see herself and her journey reflected in them. As she explains:

> One of my favorite avatars is from Neverwinter Nights. Her name is Luna, she is a warlock that is a follower of the sun god. She is what I imagine myself to be. I've wanted to be a woman for a very long time, and wanting to practice magic, wanting to learn more esoteric wisdom to help and assist the people around me, these things are how me and Luna relate to each other. Actually, Luna was a (re)creation from an old character, a male one.

Sam ended up creating another female character as well, this one an evil necromancer, as she tells us:

> In a word, I would describe her as Internal Darkness. She is everything I have yet to heal in myself, everything I have yet to let go. She is the demon I have yet to fight. I would characterize her as blind rage. . . I'm bad at letting my emotions go. I tend to bottle a lot up and anger management has always been something that I've had trouble with. Without check, the people around me could be hurt, so I would make it into something physical or "digital" so that I could have my anger in something other than myself.

Reflecting on her two favorite characters, she realizes that they are both "me" avatars, though different sides of her, like Jung's ego and shadow:

One holds the image of me toward the future and the other is what I leave behind. Luna and the necromancer were creations of my current emotional state.

As Sam goes through the process of discovering her own femininity, her female avatars become embodied expressions of her psyche. Being able to connect and identify with them has allowed Sam to work through her past "demons" and work towards actualizing the woman she wants to be.

In some ways, Sam's experiments with her characters resemble what Pearl calls interventions: Sam personally connects in variable ways to her evolving main and her various alts; these feelings of connection (or lack of connection) in turn shape the degree to which Sam derives emotional and well-being benefits from her gaming. In fact, comparing how Sam plays her *main* and her *alts* would be illuminating. However, Sam's deep gaming experience also allows us to imagine with her (in interviews) a counterfactual world where her main Luna is not female (as she is in reality) but instead male (as she might be counterfactually). And Sam tells us in no uncertain terms that such a counterfactual reality would be much less satisfying, as it would feel inauthentic and "fake." That is, she would never be able to personally connect with such a male avatar, nor successfully immerse in the game via such an avatar in ways that felt satisfying to her. Ethnography, then, provided us with opportunities to jointly imagine with Sam a counterfactual world where Sam never came to understand how her characters should be female instead of male and instead continued to play male avatars. In this alternative reality, Sam would never have created her warlock nor her necromancer. And as a result, she never would have found ways to work through her inner struggles via her characters.

Both Pearl's interventions and counterfactuals provide greater degrees of causal certainty compared to associations. However, counterfactuals, unlike interventions, allow for a more precise isolation and manipulation of a key causal factor. In this case, simply changing Luna's gender from male to female is the key to allowing Sam to connect in a deeper and more personal *me* way to her character, which in turn leads her to experience deeper forms of gaming immersion and satisfaction. Then, imagining what *might have been* if Sam had not made that simple gender change provides us with insight into how that *single* factor impacts Sam's gaming satisfaction. In our prior intervention example, by contrast, Ash's diminished personal *me* relationship to their D&D characters was inextricably bound to their changing gaming roles (from player to GM to professional GM) in ways that made it more difficult to isolate the former's causal force compared to the latter's. And likewise, simply comparing how Sam plays her *main* and her *alts* is bound up with other factors. However, ethnographically informed, counterfactual thinking also allows us to imagine the impact of potential confounders to Sam's experiences. We can ask if Sam's experiences would differ if she played less than 60 hours a week? Perhaps, but many players, who game far less, still report meaningful connections to avatars. What about Sam's age? Again, there is nothing theoretically or ethnographically to make us think that Sam's relationship to her avatar would be meaningfully different if she was 5-years older or younger. And Sam's gender? Yes—we can imagine that if Sam was not transgender, playing female avatars would not have the same effect. So, it's the combination of both Sam's and the avatar's gender that allows this experience to happen. In other words, counterfactual thinking allows us to manipulate the variables in the model to determine which drives the causal relationship.

## Representing our ethnographic findings in a graphical causal model (DAG): Clarifying causal processes and mechanisms with mixed methods research

Fig 2 presents graphically our ethnographic findings discussed in the prior sections. Our ethnography convinced us that the way players relate to their avatars can shape the quality of their

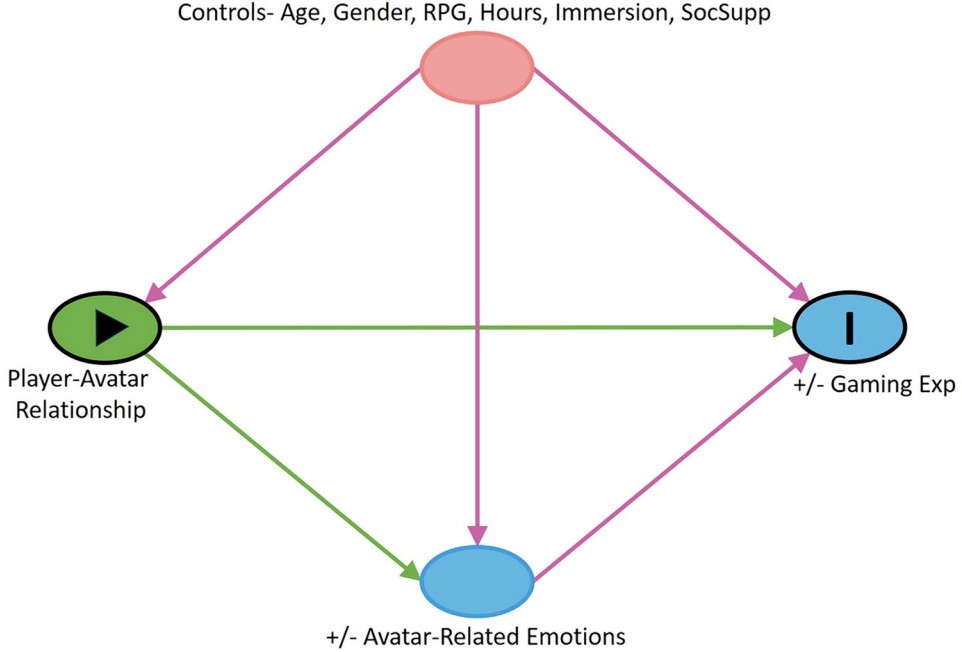

**Fig 2. Research model.**

gaming experiences and how those experiences contribute to their overall well-being. So, player-avatar relationship is represented in the model as the key predictor variable, and gaming-related well-being is the principal study outcome. That graphical model includes variables that might bias or *confound* estimates of the path between our key predictor and outcome. As revealed to a large extent in our ethnography, those variables include: gaming genre (RPGs vs. other games); propensity to have immersive experiences (both Ash and Sam highlight this); degree of involvement with gaming; social support for gaming; age; and gender. Our ethnographic research also led us to expect that distinctive emotional experiences associated with avatars would shape gaming-related well-being, including by partially *mediating* between one's relationship with their avatar (our primary predictor) and positive and negative gaming-related well-being (our outcomes). That relationship is also encoded in this graphical model.

Particularly important to causal explanation is not only *whether* a change in a predictor caused a change in an outcome but *how* and *why* a predictor effected that change. Such variables or factors that constitute the mechanism by which some cause produces an effect, as just mentioned, are commonly known as *mediators*. Connected as they are to naturally occurring events as they occur in the field, ethnographers are well-positioned to identify such factors. Of note, in social science research, mediating factors may include how study participants *interpret* and *respond to* changes in the predictor of interest [19–21, 73]. That is why understanding native points of view via long-term fieldwork—as interpretive anthropologists like Geertz remind us [10, 74]—is so important to anthropological understanding and (in our case) causal explanation. In fact, this concern with the connection between native experience and causal explanation is broader than the interpretive anthropological literature we cite here. For example, Ed Hutchins' "cognitive ethnography" [75] highlights how cultural perceptions of how the world works—what are sometimes referred to as "cultural models" [44, 45, 76]—shape human motivations in ways that direct behavior and thus should be accounted for in causal explanations [77]. And outside of anthropology, Jon Elster's influential work on "intentional

explanation" highlights the important way that motivated social action—an agent's *perception* that certain actions will lead to desirable changes in the world—importantly influences how social processes unfold in the world [37].

Relevant to this discussion on causal mediation, we also identified in our gaming research how the causal connection between player-avatar relationships and players' well-being could be mediated by distinctive emotional experiences. Via avatar-characters, roleplaying game (RPG) players experienced intense emotions, which we documented extensively in fieldnotes and explored further in interviews. Those included social and moral emotions like pride (in their characters' acts of heroism) and empathy (with their own avatars' struggles, with others' characters, and with other players), as well as other feelings like joy, wonder, hope, and gratitude. But conversely, some RPG players also reported that they could emotionally invest *too* much in avatar-characters. That could lead to too strong of a blurring of fact and fiction, intense negative and even *toxic* emotional experiences (e.g., feelings of deep shame, anger, and frustration), with the potential to disrupt their non-gaming lives. Players of other game genres could also experience similar highs and lows. For example, shooter games like *Counterstrike— Global Offensive* could be highly competitive, and sometimes felt to be "toxic" for these reasons, thus shaping players' well-being in distinctive ways, as we learned in India [78]. But overall, we found that RPG players' highs and lows were more often linked to the deeply personal relationships they formed with their avatar-characters as compared to some other cause, whereas gamers who more typically related to avatars as *objects* were more shielded from such avatar-related effects.

In the current discussion, we would emphasize that the causal processes shown graphically in Fig 2 only represent a small slice of reality: how player-avatar relationships impact gaming-related well-being. Other factors have also been shown to influence how games can come to contribute positively or negatively to players' overall well-being, including particular games' reward systems and the social context of play [78–81]. However, Fig 2 only presents the main causal relationship of interest to our study, along with any other factors (like confounders and mediators) that could affect or bias how we understand that primary causal pattern. Furthermore, Fig 2, as a *qualitative* representation of causal processes, does not say anything about the *magnitude* or *functional form* of any of the postulated causal pathways we show.

Nonetheless, representing our ethnographic findings in this way does allow the possibility of further testing with quantitative data the relationships we posit. Such tests could help us confirm our ethnographic findings and provide estimates of the magnitude of causal relationships we posit. With this in mind, and based on our ongoing ethnography, we constructed and distributed a field survey to North American gamers, and also to gamers from other parts of the world, which included: Argentina, Europe (Finland, France, and Italy), and Asia (China and Singapore) [82]. Linear regression analyses, then, allowed us to test hypotheses of relationships between different avatar relationships and gaming-related well-being, with us modeling relationships between our study predictors and outcomes using an ordinary least squares (OLS) approach [83]. To minimize bias in the estimation of the relationship between our key predictors and gaming-related well-being, we included other factors in our analysis as control variables that our ethnography and current theory suggested were likely to be causally prior influencers of both our main predictors and outcomes. For our North American sample, initial analyses reveal our survey data to be consistent with key causal patterns we postulated. For example, players reporting a more *personal* relationship with their avatars, as compared to relating to avatars as an *object*, show a substantial positive relationship to *both* positive *and* negative gaming-related well-being. (This working manuscript [83] can be made available upon request by emailing the lead author).

Here, we would emphasize that field surveys and their accompanying quantitative data and analysis, while sometimes serving as useful tools in an ethnographic toolkit [50], are not *necessary* to ethnographic analysis [51]. By this, we mean that ethnographers can use field surveys to sample relatively quickly and efficiently a broad range of respondents, though with less depth, for example, compared to qualitative interviews [60, 84]. Quantitative analysis of field survey data using techniques like regression analysis can also allow ethnographers to consider how the relationship between a predictor and outcome might change when considered alongside *multiple* confounders, *all considered simultaneously together in the analysis* [85], in ways that remain challenging in analyses of qualitative fieldnotes and interview transcripts. Further, field survey data can be analyzed via new methods in causal mediation using graphical models and counterfactual logic [29, 53, 86]. These statistical routines calculate what Pearl refers to as *natural direct effects* (DE), *natural indirect effects* (NIE), and *total effects* (TE). Pearl uses a counterfactual logic that mimics experimental research design and thinking where he examines how the relationship between a predictor and outcome varies with the mediator set at different (not actually observed and thus "counterfactual") values. For his *natural direct effects*, say, the predictor can vary between 0 (no treatment) and 1 (treatment). Pearl then examines how moving from 0 to 1 in the predictor is associated with changes in the outcome *when the mediator is set or fixed in the analysis ("set" logically and thus not actually observed) at the naturally occurring value it would have for each individual when the predictor takes on the value of 0 (the no treatment condition).* For his *natural indirect effects*, the predictor can again vary between 0 (no treatment) and 1 (treatment). In this instance, Pearl examines how moving from 0 to 1 in the predictor is associated with changes in the outcome *when the mediator is set or fixed in the analysis at the naturally occurring value it would have for each individual when the predictor takes on the value of 1 (treatment).* In Pearl's framework, the *total effect* is the sum of the natural direct and indirect effects. These new counterfactual causal mediation routines are found in Stata 18, for example [87].

Nonetheless, we have shown throughout this article how *qualitative* analysis of fieldnote and interview data can clarify in different ways the relationship between predictors and outcomes, while accounting for confounding and mediating factors. We would thus encourage ethnographers not to think that the only causal explanation worth presenting is an ironclad and standalone one, say, that accounts for *all* confounding factors in all their *multiple* combinations. Rather, ethnographers working primarily with qualitative data might instead consider how the work they do can complement and contribute to research carried out using other methods, including in team-based collaborative work [51, 72]. Indeed, they might profitably think instead how single ethnographic studies of causality—however imperfect—might be subsequently combined with and synthesized into larger "meta-ethnographies" [88–90].

## Discussion

In the current article, we have aimed to extend past classic and contemporary ethnographic work on causality by connecting to new interdisciplinary and mixed methods approaches to causal inference. In particular, we have highlighted the deeply abductive logic of ethnographic research, in reference to classic work by Peirce [23], and connecting to the work of Agar, Timmermans and Tavory, and others [24, 27]. Other contemporary ethnographers do also point to the important role of abduction in their research, but that role is generally presented as secondary to logical processes of induction (as in grounded theory) and to a lesser degree deduction (like in content analysis). For example, ethnographers can logically deduce hypotheses from current theories—be they grounded in local contexts or not—and then test (typically quantitatively) those hypotheses with new (even qualitative) data [59, 91]). Others have argued

that an abductive account both does better justice to how ethnographers arrive at certain explanations and allows for more direct advancement of current theory [28]. We have shown how this is also the case in relation to our gaming data by showing how each of our abduced propositions solved new field puzzles of ours while connecting meaningfully to current theories. Following Agar, we point to the iterative and recursive logic of this abductive process: iterative ethnographic inquiry allowed us to solve new puzzles that weren't meaningful to us until we first solved older ones, and each new proposition build upon and incorporated prior ones.

It is of course up to ethnographers themselves to decide whether a potential causal pattern is worth exploring further. Our abductive ethnographic analysis eventually led us to Proposition 2, which provided us with a potential explanation for new puzzles we encountered in the field. As we did not find in the current literature any systematic exploration of how player-avatar relationships might influence gaming-related well-being, we thought it worthwhile to continue our ethnographic inquiry along those lines. Our abductive approach helped ensure that our ever-evolving ethnography remained closely tied to current theoretical concerns in ways that might advance current understandings.

We have also highlighted how ethnographers can critically assess the relative strength of their field data to support causal inferences. Here, we draw on Pearl's ladder of causality idea and more recent research from our lab to make our points [29, 53]. We first showed how field observations pointed to associations between certain ways of relating to a gaming avatar or character and how players experienced gaming as contributing positively or negatively to their overall wellbeing. Specifically, we showed how Cindy's deeply personal connection to her gaming character was associated with both her gaming satisfaction but also her eventual distress when she was unexpectedly removed from a gaming guild she had joined. However, the potential causal relationship between player-avatar connection and gaming-related well-being in this instance is confounded by other factors like gender and social support. We then presented field observations akin to natural experiments that revealed how Ash's shift from player to GM to professional GM changed their relationship to their characters—from a personal *me-symbiote* relationship to treating characters like *others* or *objects*—which in turn eroded much of the satisfaction they derived from gaming. These ethnographic observations and interviews allowed us to ascend to Pearl's intervention ladder rung, which he and others construe as providing stronger causal evidence. For example, such evidence helps ethnographers to account for potential *reverse causality*, by showing how a given change in a postulated predictor produces another change in an outcome (rather than vice versa). Finally, we showed how Sam's gaming experiences allowed us to imagine with her alternate worlds where her main character Luna was male instead of her preferred female. Such a world was much less satisfying to Sam. This helped us pinpoint with accuracy how small factors like Luna's gender—which determined Sam's ability to personally connect with Luna as a *me-symbiote*—changed whether Sam experienced her gaming in positive or negative terms. Making sense of these shifts (with Sam's help) allowed us to ascend to Pearl's final and strongest counterfactual ladder rung: imagining what *might have been* if Luna were not female (but where other personal and social factors remain largely constant) provides us with better understanding of how a personal *me* connection to an avatar causally shapes gaming-related well-being.

In this article, space limitations have only allowed us to present a relatively limited number of ethnographic examples to illustrate our points about ethnography's potential to illuminate causal processes. In fact, both our abductive logic and ladder of causality conclusions were based on years of iterative and recursive ethnographic analysis. We have presented cases that support our points. But our actual ethnography was messier: some abduced conclusions proved unfruitful or uninteresting; and likewise, some associations, interventions, and counterfactuals supported our points about the causal connection between player-avatar

relationships and gaming-related well-being, while others did not. It is up to ethnographers to carefully weigh and compare all the accumulated evidence—from even years of observations, interviews, and lived experience—when making judgments about the causal relationship between key factors.

Likewise, we have not fully distinguished the different roles that ethnographic observations compared to interviews might play in the research discussed here, including in the development of causal explanations, though such distinctions are clearly important [60]. Thus, for example, Cindy is an active participant in her research, and her own experiences in The Sinister Smurfs guild—documented in fieldnotes—clarified for our research team potential causal connections between developing personal connections to an avatar and gaming-related well-being. Similarly, observing and documenting in fieldnotes how Ash's livelihood changes in turn impacted their relationships to their *D&D* characters (rendering those more impersonal) and subsequently their enjoyment of that game (which was decreased) clarified causal processes of interest to our study. However, discussing in interviews with Ash their experiences of those changes illuminated the emotional and other experiences that critically causally mediated between their relationships to their characters and their gaming-related well-being. Interviews also guided our understandings of Sam's experiences, though in other instances members of our research team had observed and documented in fieldnotes how their gaming characters helped them personally and other players as well work through similar gender identity issues. These examples show how ethnographic researchers collect different kinds of data [92], with interview interviews closely tied to how Ash and Sam, for example, spoke about and framed their experience to us. As such, approaches informed by ethnomethodological conversation analysis [93, 94] could usefully inform how researchers might abstract discrete causal factors from more complex conversational dynamics of those kinds, as would cognitive anthropological, hermeneutic, and linguistic considerations of how respondents *frame* and *model* their experiences to themselves and others [44, 75, 95, 96].

Overall, our analysis is different from approaches like *qualitative comparative analysis* [11], in the sense that we aim to clarify how single factors impact narrowly defined outcomes rather than how combinations of factors together as something like a gestalt might determine the shape of social life. By contrast, our ethnographic analysis in both our past and more recent work does share features with the *constant comparison* techniques of *analytic induction* and *grounded theory* [15, 17, 18]: we meticulously documented and analyzed how different ways of relating to an avatar (coded as *themes*) were associated with other factors (other themes); and we aimed to clarify through this process how certain factors (like player-avatar relationships) might causally impact others (gaming-related wellbeing). Indeed, some of those presumably largely inductive approaches explicitly incorporate a counterfactual component, as do we, by selecting cases for comparison that vary along some key dimension that allows researchers to refine causal models by ruling out alternate explanations [13, 14, 97]. Logical deduction also played a role in our analysis, e.g., in how we reasoned deductively from what we knew to theoretically sample new cases for analysis. Nevertheless, our abductive approach led us to more fully incorporate current theory into our coding and analysis process. Further, we inductively confirm our prior abduced causes in ways that give us increasing confidence in the causal processes we identified, allowing us to ascend Pearl's causal ladder from association to intervention to counterfactuals. And we also learned that fully embracing the perspectives we describe in this article involve coding in ways that are not characteristic of current grounded theory approaches. For example, ethnographers can create codes and memos related to *causes* and *effects* and *temporal priority*, as well as to potential *confounders*, *mediators*, and *colliders* that might shape or bias how they understand relationships between key variables of interest to their study. Strength of causal evidence can also be coded and memo-ed in field data

(fieldnotes and interview transcripts) as revealing *associations* between key variables of interest, or by contrast *interventions* or *counterfactual* realities. Likewise, our analysis also shares features with *mechanism* approaches to causality [19–21], given our focus in part on how third factors (like avatar-related emotions) might mediate between a primary predictor (player-avatar relationships) and outcome (gaming-related well-being).

Unlike other current qualitative approaches in sociology and related fields, our analysis of field data was also accompanied by us constantly refining causal graphical models that schematically represented our current (and ever evolving) knowledge. These models helped us clarify our thinking, for example, by focusing our attention on mediating processes and mechanisms. Ethnographers working within interpretive and other anthropological traditions often focus on how best to *interpret* (from anthropological theoretical and local cultural points of view) sociocultural realities rather than how to effectively *explain* causal processes [10, 74]. However, we have aimed to show how ethnography is well-suited to developing causal explanations. In fact, interpretive forms of evidence can be critical to developing valid causal explanations that connect more effectively to study participants' experiences and practices. This includes attending to how participants interpret and respond to the changes occurring around them, which can reveal important mediating mechanisms that connect a study's predictor and outcome. In this way of thinking, any firm distinction between interpretive and explanatory modes of analysis is unwarranted, given how respondents' interpretations of reality can critically inform the causal processes being considered.

As we have shown, the focus on DAGs also allowed us to test causal ideas developed in qualitative phases of ethnographic study with a subsequent field survey, which we analyzed quantitatively. As such, these models helped us to better integrate our different phases of research and gave us greater confidence in our causal understandings. Of note, these models, widely used in the social sciences, also allowed us to translate in a transparent fashion our research findings into a graphical form that could be used by researchers in anthropology or other fields to both scrutinize and potentially critique our work or to clarify their own research questions and hypotheses. Despite their utility, graphical models like DAGs are not currently discussed by other researchers interested in clarifying ethnography's potential to illuminate causal processes.

## Prospects

We have aimed to extend current ethnographic approaches to causality by highlighting the abductive logic of ethnography and by connecting more directly to interdisciplinary ways of assessing the strength of causal evidence and using graphical models to clarify causal patterns. Nevertheless, other topics need further clarification and explicit discussion of how they might be incorporated into current ethnographic approaches to explaining causal processes. These include: the role that logical induction and deduction play in relation to abduction in ethnographic analysis; the extent that induction, deduction, and abduction approaches are *necessary* or *sufficient* in the logical definitional sense for developing causal ethnographic arguments; how ethnographic data might be coded and analyzed to clarify potential moderation effects, where one factor changes the relationship between other factors; greater attentiveness to selection bias and collider effects in ethnographic data; using ethnography to develop culturally valid measures that can be employed in causal studies; integrating the current approach with field experiments or longitudinal data; more refinement of ethnographic analysis of interventions and counterfactuals, say, by documenting ethnographically naturally occurring experiments or developing interview protocols to explore more effectively counterfactual realities; attention to how different kinds of ethnographic data—say, observations versus interviews—differently inform causal analysis.

## Conclusion

Classic ethnographic accounts illuminated causal processes, but they were not overly reflexive on exactly how this occurred or how a causal explanation might differ from some other kind of account. By contrast, Geertz argued (convincingly to many) that ethnography is better suited to help a researcher *interpret* (through cultural outsider *etic* and insider *emic* frameworks of meaning) rather than *causally explain* social reality. In our own approach, we highlight how ethnographers can use abductive logic (combined with logical induction and deduction), strength of evidence assessments, and graphical models to help solve cultural puzzles and build convincing causal arguments that contribute to current theory. This includes demonstrating how causal mediation analysis, facilitated by graphical modeling, can clarify how culturally informed motives and "native points of view" shape distinctive behaviors and experiences. It is more commonly understood that ethnography might contribute to causal understandings by helping to generate hypotheses, by providing contextual understandings of causal processes, by theory building, and by providing complementary qualitative data that might inform subsequent quantitative tests of causal processes. We agree with all those points; however, we build a stronger case here that ethnographic analysis can itself provide the basis for making convincing causal arguments. Further, any quantitative analyses emerging from prior qualitative data and analysis are fully ethnographic, if they remain tied to local sociocultural realities.

Our account of ethnography is consistent, we believe, with earlier classic approaches to ethnography, and with key assumptions of interpretive and other contemporary more inductive grounded theory approaches to ethnography as well. For example, following interpretive anthropological tenets, we recognize that deep immersion in cultural settings is necessary to the analytical processes we describe, as is grasping local points of view and appreciating sociocultural complexity. And as we have partially shown, analytical induction, akin to grounded theory analysis, helps ethnographers validate abduced propositions through a constant comparative method. In fact, distinguishing interpretive from explanatory approaches to ethnography rests in part on a false dichotomy: native points of view on specific cultural objects and processes—and their culturally-driven behaviors as well—can be determinative and thus predictive factors in a causal model (or by contrast posited as outcomes); and asking cultural natives to imagine how cultural processes they know might play out differently in other circumstances can provide ethnographers with insights into counterfactual realities and thus potentially help ethnographers sharpen their causal inferences. Nevertheless, we think that ethnographers could further improve, or reconsider, their understanding of causal processes by following some of the principles we describe. That is, ethnographers could profitably learn from other fields where attention to logical abduction, strength of evidence assessments, and graphical models like we describe are commonly employed. Likewise, we hope that our approach and analysis might help researchers in other fields to better appreciate the great potential ethnographic research has for clarifying causal sociocultural processes.

## Supporting information

**S1 Appendix. Causal concepts primer.**
(DOCX)

## Acknowledgments

We thank the many gamers who have participated in our gaming research in its various phases.

## Author Contributions

**Conceptualization:** Jeffrey G. Snodgrass, H. J. François Dengah, II, Seth I. Sagstetter, Katya Xinyi Zhao.

**Data curation:** Jeffrey G. Snodgrass.

**Formal analysis:** Jeffrey G. Snodgrass.

**Funding acquisition:** Jeffrey G. Snodgrass.

**Investigation:** Jeffrey G. Snodgrass, H. J. François Dengah, II, Seth I. Sagstetter, Katya Xinyi Zhao.

**Methodology:** Jeffrey G. Snodgrass, H. J. François Dengah, II.

**Project administration:** Jeffrey G. Snodgrass.

**Resources:** Jeffrey G. Snodgrass.

**Supervision:** Jeffrey G. Snodgrass.

**Visualization:** Jeffrey G. Snodgrass, H. J. François Dengah, II.

**Writing – original draft:** Jeffrey G. Snodgrass, H. J. François Dengah, II, Seth I. Sagstetter, Katya Xinyi Zhao.

**Writing – review & editing:** Jeffrey G. Snodgrass, H. J. François Dengah, II, Seth I. Sagstetter, Katya Xinyi Zhao.

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
