## [Decision Letter · Decision Letter 0]

14 Mar 2024

PONE-D-24-00708Causal Inference in Ethnographic Research: Refining Explanations with Abductive Logic, Strength of Evidence Assessments, and Graphical ModelsPLOS ONE

Dear Dr. Snodgrass,

Thank you for submitting your manuscript to PLOS ONE. After careful consideration, we feel that it has merit but does not fully meet PLOS ONE’s publication criteria as it currently stands. Therefore, we invite you to submit a revised version of the manuscript that addresses the points raised during the review process.

**Both reviewers raised some interesting points and were generally favorable towards your work. I would invite you to address their comments in the manuscript. **

We look forward to receiving your revised manuscript.

Kind regards,

Stefaan Six, Ph.D.

Academic Editor

PLOS ONE

Journal Requirements:

The Foundation for Psychocultural Research (https://thefpr.org/). “Online gaming involvement, avatar identification, and emotion regulation in five culture areas: A multi-level cultural norm and social network approach.” P.I.: Jeffrey G. Snodgrass.

Reviewers' comments:

Reviewer's Responses to Questions

**Comments to the Author**

1. Is the manuscript technically sound, and do the data support the conclusions?

Reviewer #1: Yes

Reviewer #2: Yes

2. Has the statistical analysis been performed appropriately and rigorously? 

Reviewer #1: N/A

Reviewer #2: N/A

3. Have the authors made all data underlying the findings in their manuscript fully available?

Reviewer #1: Yes

Reviewer #2: Yes

4. Is the manuscript presented in an intelligible fashion and written in standard English?

Reviewer #1: Yes

Reviewer #2: Yes

5. Review Comments to the Author

Reviewer #1: I found the paper to be clearly written, well paced, and a very interesting read. The aims, structure and statement of contribution all make sense. If anything the paper may be a little on the long side, with key messages reiterated multiple times. For me this was a strength but I’m not sure how it sits with any journal requirements in terms of manuscript length. The sections presenting examples from the authors’ own work are excellent.

My comments (below) are either speculative or very minor and are not offered as conditions of acceptance as I think the paper is already substantive in contribution and pretty close to being ‘publication-ready’.

comments:

In the early sections it would be interesting to know if ethnographers from the interpretivist school have been actively hostile to these kinds of approaches. Was this ever a battleground in the ‘paradigm wars’? I confess I was unaware of this other tradition within ethnographic research.

Linked to this, I’d be interested in clarification of the intended relationship to quantitative analysis. For example, does the proposed approach suggest a logic whereby most/all qualitative research is followed by quantitative analysis of the causal propositions generated? (in service of for example developing theories with predictive power). Or would conclusions based purely on qualitative data suffice in some instances? This is touched upon in the paper but only briefly. Linked to this, do the authors have a view on meta-ethnography as a means of establishing causality based on pooled qualitative data?

Minor points -

p. 1 consider rephrasing: ‘ethnography is presented as not seem well-suited to’ (page 1) and ‘Theoretical sampling of next cases is critical in the manner is allows researchers’ (p.2)

p.2 ‘manipulating’ seems a strange choice of words. Maybe ‘examining’?

p.20 change ‘before long’ to ‘long before’

The authors could review when and how often italics are used – e.g. interventions is possibly italicised beyond the point where this is necessary

Clarify how Peirce is being cited – a source appears as 45 in the reference list but this isn’t always cited when Peirce is referred to through the text (including the first instance)

Reviewer #2: The authors of this article describe a path towards a more epistemically diverse, and scientifically rich, landscape of ethnographic research. According to this vision, ethnographic analysis should not merely aim to provide inductive accounts of social phenomena but can and should be explicitly oriented towards making and probing claims about causality. In making the case for causal claims as a legitimate epistemic goal for ethnography, the authors turn towards recent developments in the philosophy of social science around 'abductive analysis' and analytical tools for exploring causality, particularly adopting Pearl's ‘ladder of causality’. The latter is a causal hierarchy spanning from association (‘what is’), via intervention (‘what if’) to counterfactuals (‘why'). Their arguments are supported with examples from ongoing field research on games, play and avatar identities in virtual and non-virtual worlds. These examples are relevant (although based on a particular slice of ethnographic data, as I mention below) and applied in a way that nicely illustrates the claims being made.

In sum, the authors present an enticing vision for ethnography that I wholeheartedly embrace, and I applaud the authors for their contribution. This work should be welcomed by anthropologists and other social scientists that grapple with ethnographic methods to study the social nature of meaning. This include those scholars who primarily approach social life through the lens of qualitative methods (under the label of ‘inductive’ or ‘explorative’ social science), but the article should be of particular interest among the growing group of anthropologists and ethnographers with a naturalistic ambition for inquiry into social meaning and practice.

Ideally, one would hope that this paper would see publication in one of the top anthropology journals, but I also understand the author's motives for publishing this solid piece in a more generalist journal like PLOS One, and for making it accessible to a broader group of readers who are drawing on ethnographic methods.

Here are two larger issues that I would like to see addressed (although I leave it to the authors to operationalize this within the constraints of their text, considering the other review comments):

Major point 1)

• The examples taken from the authors’ own ethnography are all primarily interview- based. This leads me to ask: does the argument impact the variety of phenomena or ethnographic data that can be accommodated or explained through the principles of abductive inference and the levels of causality-approach? For instance, there may be differences between how interview materials and observational materials can be used for causal inference within this framework. I therefore wonder whether this framework for telling causal stories through ethnographic data is somehow constrained by the nature of ethnographic representations.

To elaborate: there is likely some important differences between how ethnographic field reports based on descriptive (propositions) and non-descriptive representations (reproductions and interpretations) can be recruited for theorizing and causal storytelling (this goes back to Dan Sperber’s classic formulation in his chapter Interpretative ethnography and theoretical anthropology, ‘On anthropological knowledge’1985). Ethnomethodological conversation analysis, for example, have long recognized that examining the nature of social phenomena may benefit from very detailed (micro)analysis of practices at the scale of conversational time, and have developed notational systems for capturing various aspects of multimodal action. Also, the body of work done under the label of what Ed Hutchins calls "cognitive ethnography" has similar epistemic aims as those promoted here, often involving causal claims about the nature of cognitive-cultural systems involving materiality and human action, and often rely on particular types of ethnographic representations to make (causal) claims about the cognitive nature of cultural practices.

Accordingly, are there representational constraints on the ethnographic data when used for causal analysis in the ways suggested by the authors?

Major point 2)

• The background is informative, and it builds on relevant references, at the cutting edge of inferential questions in qualitative social science (with the above exception). Still, in its treatment of a causal project for social analysis based on ethnography, the paper appears to have overlooked an important body of work in the philosophy of social science that speak directly to the sort of explanatory models promoted by the authors, namely Jon Elster's work on what he calls ‘intentional explanations’. These are basic explanatory building blocks in the social sciences (some would say that they are the distinguishing characteristic differentiating social from natural science), and they are fundamentally describing the meaning of a given (social) action. For Elster, intentional explanations comprise a central explanatory logic that is crucial for all the human sciences that we may dub as "hermeneutic", which includes those relying on ethnographic data. In its basic form, an intentional explanation of an action X consists in showing that it – according to the agent’s perception – was the best means to realize her desires. Additionally, this correspondence between action, desires, and perceptions was not due to chance, but because it was adapted to the agent's perceptions and desires. Logically, these accounts take the ideal form: Person X intends to achieve Y. Person X believes that action A is the best means to achieve Y. Therefore, person X performs action A.

Although Elster logically distinguishes this class of intentional explanations from causal explanations, and functional explanations (I believe the authors of this paper are less concerned about such distinctions), I think it is worth mentioning and clarifying how you position yourself with respect to Elster's position, in the context of this paper (given the influential position his account has). My key point here is that the sort of hermeneutic analysis that relies on the logic described above, should be fundamentally regarded as a type of explanation, too. An overview of this position can be found in Elster’s seminal ‘Explaining technical change’ (chapter 3, as well as later works).

One additional reason for making this point, is that Elster's view is actually quite closely aligned with the work on social mechanisms that the authors cite on page 3 (reference 19, such as Hedstrom and Svedberg). I get the impression that the authors too believe that accounting for social mechanisms is a legitimate pursuit for social scientist in the ethnographic tradition, but this could be clarified.

A few minor issues:

1) Generally, the paper is well-written, clearly articulated, and easy to follow.. However, sentence on page 17 should be revised for clarity: "Deduction also has its place: to gain greater confidence in an abduced explanation, ethnographers can test whether other propositions logically deduced from the data are also consistent with the collected ethnographic data."

2) Is it possible to get higher resolution figures? The resolution is a bit low.

3) 'Strength of evidence assessments' plays a key role in the title but does not get that much attention in the paper, as that section is a bit shorter than the others. This is not a problem, but perhaps consider whether to keep this element in the title?

6. PLOS authors have the option to publish the peer review history of their article (what does this mean?). If published, this will include your full peer review and any attached files.

Reviewer #1: **Yes: **Iestyn Williams

Reviewer #2: **Yes: **Mads Solberg

---

## [Author Response · Author response to Decision Letter 0]

2 Apr 2024

(Note: We also uploaded these comments as a supporting file.)

Response to Reviewers

PONE-D-24-00708

Causal Inference in Ethnographic Research: Refining Explanations with Abductive Logic, Strength of Evidence Assessments, and Graphical Models

Reviewers' comments:

Reviewer's Responses to Questions

Comments to the Author

1. Is the manuscript technically sound, and do the data support the conclusions?

Reviewer #1: Yes

Reviewer #2: Yes

2. Has the statistical analysis been performed appropriately and rigorously?

Reviewer #1: N/A

Reviewer #2: N/A

3. Have the authors made all data underlying the findings in their manuscript fully available?

Reviewer #1: Yes

Reviewer #2: Yes

4. Is the manuscript presented in an intelligible fashion and written in standard English?

Reviewer #1: Yes

Reviewer #2: Yes

5. Review Comments to the Author

Reviewer #1: I found the paper to be clearly written, well paced, and a very interesting read. The aims, structure and statement of contribution all make sense. If anything the paper may be a little on the long side, with key messages reiterated multiple times. For me this was a strength but I’m not sure how it sits with any journal requirements in terms of manuscript length. The sections presenting examples from the authors’ own work are excellent.

My comments (below) are either speculative or very minor and are not offered as conditions of acceptance as I think the paper is already substantive in contribution and pretty close to being ‘publication-ready’.

We appreciate your favorable assessment of our work and the helpful comments you provide below. We have added text to respond to your comments and those of the other reviewer. The body of the main text remains at ~13,600 words, which is in line with other leading journals in our field like Current Anthropology (12,000 word limit on main text plus ~2000 words additionally to respond to reviewer comments). We agree with you that even certain repeated passages help with flow and understanding of key ideas. We are not aware of any PLOS One word limits. 

comments:

In the early sections it would be interesting to know if ethnographers from the interpretivist school have been actively hostile to these kinds of approaches. Was this ever a battleground in the ‘paradigm wars’? I confess I was unaware of this other tradition within ethnographic research.

We provide a new quote (p.2) by Geertz that shows his opposition to causal analysis, along with a new paragraph at the end of our Introduction referencing anthropology’s paradigm or science wars (p. 5). However, as you see, we aim to strike a conciliatory tone in this new paragraph. This is because even anthropologists inspired by various dimensions of Geertz’s interpretive anthropology (including ourselves) take different stances toward causal analysis, and so we hesitate to characterize such approaches in too broad of brush strokes. Further, we hope to build bridges to scholars working in more humanistic, critical, and interpretive schools by pointing to the utility to such approaches of the analytical strategies we describe in our article. 

Linked to this, I’d be interested in clarification of the intended relationship to quantitative analysis. For example, does the proposed approach suggest a logic whereby most/all qualitative research is followed by quantitative analysis of the causal propositions generated? (in service of for example developing theories with predictive power). Or would conclusions based purely on qualitative data suffice in some instances? This is touched upon in the paper but only briefly. Linked to this, do the authors have a view on meta-ethnography as a means of establishing causality based on pooled qualitative data?

We added a new paragraph to respond to these important questions (pp. 35-36, the concluding paragraph of that section). There, we highlight the utility for ethnographic research of quantitative analysis of field survey data, e.g., expanding a sample, simultaneous consideration of multiple confounding factors, and new frontier approaches by Pearl and others to counterfactual mediation analysis. However, we also emphasize that though quantitative approaches can usefully extend and complement quantitative data and analysis they are not necessary to ethnographic studies. (If you’re interested, see our In Press article on this topic (Reference #51).) Note that we also now cite in this new paragraph work by Agar and others on “meta-ethnography” and “meta-analyses” of social science research. Thank you for these comments, which allowed us to clarify and expand for readers our position on these important issues. 

Minor points -

p. 1 consider rephrasing: ‘ethnography is presented as not seem well-suited to’ (page 1) and ‘Theoretical sampling of next cases is critical in the manner is allows researchers’ (p.2)

Corrected by removing “seem.”

p.2 ‘manipulating’ seems a strange choice of words. Maybe ‘examining’?

“Manipulating” changed to “examining.” (Our idea was that ethnography does not lend itself to experimental manipulation of a treatment factor, but we see how that could be confusing.)

p.20 change ‘before long’ to ‘long before’

Changed.

The authors could review when and how often italics are used – e.g. interventions is possibly italicised beyond the point where this is necessary

We’ve reduced our use of italics throughout. 

Clarify how Peirce is being cited – a source appears as 45 in the reference list but this isn’t always cited when Peirce is referred to through the text (including the first instance)

We now cite Peirce more consistently, though we also try to make it clear that we draw certain ideas from contemporary social science interpretations of Peirce by scholars such as Agar, Timmermans, and Tavory.

Thank you again for your thoughtful comments that helped us clarify and expand certain points and overall elevate the quality of our work. 

Reviewer #2: The authors of this article describe a path towards a more epistemically diverse, and scientifically rich, landscape of ethnographic research. According to this vision, ethnographic analysis should not merely aim to provide inductive accounts of social phenomena but can and should be explicitly oriented towards making and probing claims about causality. In making the case for causal claims as a legitimate epistemic goal for ethnography, the authors turn towards recent developments in the philosophy of social science around 'abductive analysis' and analytical tools for exploring causality, particularly adopting Pearl's ‘ladder of causality’. The latter is a causal hierarchy spanning from association (‘what is’), via intervention (‘what if’) to counterfactuals (‘why'). Their arguments are supported with examples from ongoing field research on games, play and avatar identities in virtual and non-virtual worlds. These examples are relevant (although based on a particular slice of ethnographic data, as I mention below) and applied in a way that nicely illustrates the claims being made.

In sum, the authors present an enticing vision for ethnography that I wholeheartedly embrace, and I applaud the authors for their contribution. This work should be welcomed by anthropologists and other social scientists that grapple with ethnographic methods to study the social nature of meaning. This include those scholars who primarily approach social life through the lens of qualitative methods (under the label of ‘inductive’ or ‘explorative’ social science), but the article should be of particular interest among the growing group of anthropologists and ethnographers with a naturalistic ambition for inquiry into social meaning and practice.

Ideally, one would hope that this paper would see publication in one of the top anthropology journals, but I also understand the author's motives for publishing this solid piece in a more generalist journal like PLOS One, and for making it accessible to a broader group of readers who are drawing on ethnographic methods.

Thank you for your positive assessment of our work, which we appreciate. 

Here are two larger issues that I would like to see addressed (although I leave it to the authors to operationalize this within the constraints of their text, considering the other review comments):

Major point 1)

• The examples taken from the authors’ own ethnography are all primarily interview- based. This leads me to ask: does the argument impact the variety of phenomena or ethnographic data that can be accommodated or explained through the principles of abductive inference and the levels of causality-approach? For instance, there may be differences between how interview materials and observational materials can be used for causal inference within this framework. I therefore wonder whether this framework for telling causal stories through ethnographic data is somehow constrained by the nature of ethnographic representations.

We have edited our manuscript to clarify that some of our data comes from ethnographic observations documented in fieldnotes, including of our research team members’ own experiences (e.g., see Cindy’s vignette, pp. 20-3). 

Your comment raises important issues that we cannot fully respond to within the constraints of this article. However, we address with new writing (e.g., a substantially reworked graphical models findings section, pp. 32-6) one substantial issue related to causal mediation analysis, which also allows us to respond to your second major point (discussed further below). In that new writing, we emphasize the usefulness of interview data to illuminate not only whether a change in a predictor is associated with a change in an outcome but the causal mechanisms and processes potentially explaining how and why change occurs. This is because certain mediating factors can include how study participants interpret and respond to changes occurring around them, which can be usefully clarified in the context of interviews. So, field observations can help clarify how changes in a predictor relate to changes in an outcome, as can interviews, but interviews are particularly useful to explaining how certain such causal connections occur via respondents’ experiences. 

For these clarifications and expansions, see especially the new paragraph beginning on p. 33 in the expanded section on causal mediation (“Particularly important to causal explanation…”), where you’ll also see we now also discuss the work of Elster, Hutchins, and others (more on that in a moment). See too the subsequent new paragraph (p. 34) where we present ethnographic material on causal mediation in the context of our gaming research (“Relevant to this discussion on causal mediation, …”). And we also highlight differences between observational and interview data in our gaming research in a new substantial paragraph (pp. 39-40) in the Discussion (“Likewise, we have not fully distinguished…”). There, we again point to the importance of interview data in particular for social scientific analysis of causal mediation. Finally, we reference this important issue in a new line in the Prospects section.

To elaborate: there is likely some important differences between how ethnographic field reports based on descriptive (propositions) and non-descriptive representations (reproductions and interpretations) can be recruited for theorizing and causal storytelling (this goes back to Dan Sperber’s classic formulation in his chapter Interpretative ethnography and theoretical anthropology, ‘On anthropological knowledge’1985). Ethnomethodological conversation analysis, for example, have long recognized that examining the nature of social phenomena may benefit from very detailed (micro)analysis of practices at the scale of conversational time, and have developed notational systems for capturing various aspects of multimodal action. Also, the body of work done under the label of what Ed Hutchins calls "cognitive ethnography" has similar epistemic aims as those promoted here, often involving causal claims about the nature of cognitive-cultural systems involving materiality and human action, and often rely on particular types of ethnographic representations to make (causal) claims about the cognitive nature of cultural practices.

Accordingly, are there representational constraints on the ethnographic data when used for causal analysis in the ways suggested by the authors?

Yes, these are important points, and we address them in part via our new discussion of causal mediation analysis in particular, as we say above. Note our new references to the work of Ed Hutchins and other cognitive anthropologists (p. 33) and to Sperber, ethnomethodologists, and others in our Discussion (pp. 39-40). 

Major point 2)

• The background is informative, and it builds on relevant references, at the cutting edge of inferential questions in qualitative social science (with the above exception). Still, in its treatment of a causal project for social analysis based on ethnography, the paper appears to have overlooked an important body of work in the philosophy of social science that speak directly to the sort of explanatory models promoted by the authors, namely Jon Elster's work on what he calls ‘intentional explanations’. These are basic explanatory building blocks in the social sciences (some would say that they are the distinguishing characteristic differentiating social from natural science), and they are fundamentally describing the meaning of a given (social) action. For Elster, intentional explanations comprise a central explanatory logic that is crucial for all the human sciences that we may dub as "hermeneutic", which includes those relying on ethnographic data. In its basic form, an intentional explanation of an action X consists in showing that it – according to the agent’s perception – was the best means to realize her desires. Additionally, this correspondence between action, desires, and perceptions was not due to chance, but because it was adapted to the agent's perceptions and desires. Logically, these accounts take the ideal form: Person X intends to achieve Y. Person X believes that action A is the best means to achieve Y. Therefore, person X performs action A.

Although Elster logically distinguishes this class of intentional explanations from causal explanations, and functional explanations (I believe the authors of this paper are less concerned about such distinctions), I think it is worth mentioning and clarifying how you position yourself with respect to Elster's position, in the context of this paper (given the influenti

---

## [Editor Report · Decision Letter 1]

11 Apr 2024

Causal inference in ethnographic research: Refining explanations with abductive logic, strength of evidence assessments, and graphical models

PONE-D-24-00708R1

Dear Dr. Snodgrass,

We’re pleased to inform you that your manuscript has been judged scientifically suitable for publication and will be formally accepted for publication once it meets all outstanding technical requirements.

Kind regards,

Stefaan Six, Ph.D.

Academic Editor

PLOS ONE